# How Self-Control Predicts Moral Decision Making: An Exploratory Study on Healthy Participants

**DOI:** 10.3390/ijerph18073840

**Published:** 2021-04-06

**Authors:** Chiara Lucifora, Gabriella Martino, Anna Curcuruto, Mohammad Ali Salehinejad, Carmelo Mario Vicario

**Affiliations:** 1Department of Cognitive Science, University of Messina, 98122 Messina, Italy; chiara.lucifora@unime.it (C.L.); annacurcuruto94@gmail.com (A.C.); 2Department of Clinical and Experimental Medicine, University of Messina, 98122 Messina, Italy; gabriella.martino@unime.it; 3Leibniz Research Centre for Working Environment and Human Factors, Department of Psychology and Neurosciences, 44139 Dortmund, Germany; salehinejad@ifado.de

**Keywords:** compulsivity, filler dilemmas, impulsivity, incidental dilemmas, inhibitory control, instrumental dilemmas, moral dilemmas

## Abstract

Research on moral reasoning calls into question self-control, which encompasses impulsivity, compulsivity, and inhibitory control. However, a thorough investigation exploring how these three dimensions can affect moral reasoning in response to different scenarios is unavailable. We addressed this topic by testing the predictive role of these three dimensions of self-control on appraisals for ethical violations related with different types of scenarios. Overall, our results suggest that all three dimensions of self-control are involved in moral reasoning, depending on the type of appraisal and provided moral scenarios.

## 1. Introduction

Moral behavior can be intended as the attitude to respect social norms, while immoral behavior is the aptitude to violate these norms [1].

The research in social and psychological science has shown that moral behavior can be influenced by several socioenvironmental variables, including cultural values and traditions e.g., [2], emotions such as disgust sensitivity and anger [3,4,5,6], motivational states such as appetite [7,8], and executive functions such as self-control [9]. For example, Fuijita [10] has suggested that the inhibition of impulses and the ability to resist temptation have a great weight on the moral process; moreover, self-control can help encourage the right behavior and inhibit the wrong one [11].

Self-control is an executive function [12] known to include impulsivity, compulsivity and inhibitory control [13]. Impulsivity, described as a predisposition to rapid or unplanned reactions without considering the possible negative consequences [14] is an individual trait that might affect moral decision-making. For instance, the recent study of Ju et al. [15] shows that individuals with a high level of impulsivity are disinterested in the consequences of their actions, and they are more able to harm other people to reach their goals when dealing with extreme, simulated accident situations. Further insights on the influence of impulsivity on morality are provided by research linking impulsivity with sexual assault, especially in the domestic context e.g., [16,17,18,19,20].

Compulsivity refers to repetitive acts not in line with one’s overall goal, which are performed in the absence of awareness [21]. Compulsivity was recently indicated by Grubbs et al. [22] as a predictor of moral disapproval in line with the demonstration of a covariation between compulsive sexual behavior, moral disapproval, and the use of pornography.

Inhibitory control, the ability to suppress or countermand thoughts, actions, or feelings [23], has been associated with morality in antisocial personality disorder e.g., [24]. Furthermore, a recent work suggested the importance of potentiating inhibitory control to improve moral behavior [25].

The study of clinical populations provides a further support to the link between self-control and morality. The Obsessive-Compulsive Disorder (OCD), a clinical condition characterized by high impulsivity, compulsivity e.g. [26], and fear of losing control, is affected by high sensitivity to ethical violations. For example, the recent study by Hosseinzadeh et al. [27] shows that patients with OCD have more rigid and hypersensitive moral judgments, which was explained in relation to their impaired inhibitory control.

In general, the literature examined above supports the relevance of these three dimensions of self-control in moral decision-making. However, knowledge of their specific contribution on different types of moral reasoning is missing. The Padua Inventory (PI) [28,29] and a small set of moral dilemmas made by Lotto et al. [30] were used to address this gap of knowledge. Details on the core elements associated with PI and the selected dilemmas are provided in the methods section.

Our research hypothesis is that impulsivity, compulsivity, and inhibitory control may predict moral decision-making based on the type of scenario and the type of appraisal in response to these scenarios.

## 2. Materials and Method

### 2.1. Participants

Our study involved a total of 107 subjects (mean age = 27.06 ± 7.90 SD, the age range was 20–60 years old), consisting of 33 males, 73 females. One participant did not declare the gender. Of the total, 31 participants were workers, 66 participants were students, and 10 participants were unemployed. Most of the participants were students of the Department of Cognitive, Psychological, Pedagogical and Cultural Studies of the University of Messina, where the proportion of females is higher than the proportion of males. Because of restrictions related to the COVID pandemic, participants were recruited through social platforms such as Facebook, WhatsApp, and personal email. No gratuity was given to the participants and no time limits were given to complete the online survey. All the participants gave their consent to participate in the research. The study was approved by the local ethics committee (COSPECS Department, University of Messina).

### 2.2. Instruments

The PI [28] includes 60 items, aiming at identifying the presence of obsessive thoughts and impulses, as well as compulsive behaviors, in clinical and nonclinical populations. Each item is associated with a 5-point Likert scale that investigates the sense of discomfort (from 0 = nothing to 4 = a lot) and includes 4 subscales: (i) impaired control of mental activities; (ii) becoming contaminated; (iii) checking behaviors; (iv) urges and worries of losing control over motor behaviors [28]. This instrument allows assessment, by means of a single instrument, of impulsivity (via urges and worries of losing control over motor behaviors subscale); compulsivity (via checking behaviors and becoming contaminated subscales); inhibitory control (via impaired control of mental activities subscale). This latter dimension seems to share a lot of similarities with the theory of moral disengagement [31], which highlight the role of self-regulation mechanisms on morality.

In the first subscale (i.e., impaired control of mental activities) the questions investigate the ability to manage (i.e., suppress) unpleasant thoughts (e.g., ruminations), and doubts about one’s responsibility in everyday events. This subscale allows testing of the inhibitory control ability of participants. Typical items of this subscale are “I have a hard time making decisions even for minor things” or “I have doubts and problems about a lot of the things I do”.

The second subscale (i.e., becoming contaminated) investigates the fear of being mentally or physically contaminated and the consequent (compulsive) cleaning acts, for example “I feel my hands dirty when I touch the money” or “I avoid using the public telephone because I am afraid of disease and contamination”; the third subscale (i.e., checking behavior) refers to compulsive behaviors such as having to repeat the same action several times, for example, “I go back to check the gas or water taps after having closed them” or “when I use the money I count it several times”; the fourth subscale (i.e., urges and worries of losing control over motor behaviors) refers to impulsive behaviors such as restless thoughts at the sight of weapons or the need to break objects for no reason, for example, “watching an approaching train I happen to think that I might throw myself on the tracks” or “when I drive I happen to feel the urge to run over something or someone”; the total PI score indicates whether the participant can be classified as suffering from OCD. To total Cronbach’s alpha score is, 0.94, ranging from 0.70 to 0.90 for respective subscales.

A set of moral dilemmas [30] was used to explore the role of the individual variables mentioned above on moral behavior. The original set [30] includes 75 dilemmas, divided into three groups:Incidental dilemmasInstrumental dilemmasFiller dilemmas

In incidental dilemmas, the sacrifice of the person is an expected but unwanted consequence of an action aimed at saving a greater number of people; in the instrumental dilemmas, the person who is sacrificed is used to save more people; in both categories, participants were dealing with situations implying (self-involvement) or not implying (others-involvement) a personal involvement. Dilemmas with self-involvement require participants to make a choice to save themselves and other people; dilemmas with involvement of others require participants to make a choice about the death of other people. A typical incidental dilemma provides scenarios similar to that described in the Trolley Dilemma [32], in which the participant is asked to move an object (such as press a button or moving a lever) to kill or save other people; A typical instrumental dilemma provides scenarios similar to that described in the typical Footbridge Dilemma [33], in which participants have to use a subject’s body to save other people. In filler dilemmas. scenarios are described about morally inappropriate actions such as stealing, lying, and being dishonest, but never imply the killing [30]. For our purpose we use only 15 dilemmas, consisting of 6 incidental dilemmas (3 with self-involvement and 3 with others-involvement); 6 instrumental dilemmas (3 with self-involvement and 3 with others-involvement); and 3 filler dilemmas.

Specifically, for the incidental dilemmas with others-involvement we use “hospital”, “nurse”, “quarantine”; with self-involvement we use “nuclear power plant”, “window”, “bodyguard”. For the instrumental dilemmas with others-involvement, we use “door”, “transplant”, “vitamins”; with self-involvement we use “helicopter”, “Jeep”, “kidnapping and escape”. For the filler dilemmas we use “charity”, “supermarket” and “wallet” (see supplemental materials for further details about the adopted dilemmas).

Each dilemma is associated with questions exploring the moral acceptability via an 8-point scale (0 = not at all, 7 = completely); the moral valence (i.e., the degree of pleasantness/unpleasantness) via a 9-point scale (1 = dislike, 9 = like); and the arousal (i.e., the degree of calm/activation) via a 9-point scale (1 = calm, 9 = activation). Acceptability and valence can be considered indices to explore the explicit appraisal (i.e., what people say) of participants in response to the presented moral scenarios; arousal can be considered an index to explore the implicit—i.e., affective-appraisal (i.e., what people feel) in response to the presented moral scenarios. The latter suggestion is in line with the evidence of distinct moral processing under different levels of emotional arousal [34]. Moreover, the study of Lang et al. [35] has found a significant covariation between affective valence judgment and arousal ratings.

### 2.3. Procedure

All these questionnaires were administered to the participants remotely, through a “Google forms” platform. First, we gathered consent to participate in the study, and demographic variables such as age, gender, and occupation. Next, the participants were asked to complete the PI and provide responses to the proposed dilemmas.

## 3. Data Analysis and Results

First, we explored any role of demographics (age, sex, occupation) in predicting appraisals for the provided dilemmas. Next, to explore any relationship between the scores associated with the four PI subscales and moral decision-making for the different types of scenarios, we planned to perform correlation analyses. Since no normal distribution of the scores was detected (Shapiro–Wilk *p* < 0.011) we used the Spearman correlation test. To confront differences among the type of occupations, we used repeated measures ANOVA. Student’s *t*-test was used to compare any difference between male and female participants. The *p*-level was set at 0.05. A detailed report of significant results is included in the tables provided below.

### 3.1. Instruments

The Table 1 provides means and standard deviations of scores related with PI and the adopted moral dilemmas.

### 3.2. Demographics

No correlations were documented between age and ratings provided for acceptability (r = −0.146; *p* = 0.133), valence (r = 0.045; *p* = 0.653) and arousal (r = −0.075; *p* = 0.439) appraisals.

A significant difference was reported between male (M = 5240) and female (M = 6020) participants for arousal score (t(104) = 2118; *p* = 0.036). No significant differences were found for acceptability (t(104) = 0.154; *p* = 0.877) and valence (t(104) = 0.712; *p* = 0.477) scores when comparing male with female participants. Finally, the ANOVA did not detect a significant main effect of the occupation variable [F(2, 104) = 0.061, *p* = 0.940], as well as for the occupation x type of appraisal interaction term [F(4, 208) = 0.059, *p* = 0.993]. The main effect of the appraisal variable was significant [F(2, 208) = 5193, *p* < 0.001].

### 3.3. Relationship between Incidental Moral Dilemmas and PI Scores

The analysis documents several positive correlations. First, we found two correlations between becoming contaminated and implicit appraisal (i.e., arousal) for ethical violations involving self and others. We also found a positive correlation between checking behavior and explicit appraisal (i.e., valence) for ethical violations. Therefore, the higher the compulsivity the higher the valence and the arousal scores in response to ethical violations.

Next, we found significant correlations between urges and worries of losing control over motor behaviors and respective explicit appraisals such as valence and acceptability ratings in the context of self-involvement. Therefore, the higher the impulsivity the higher the degree of perceived pleasure and acceptability of ethical violations. Moreover, impulsivity positively correlates with acceptability scores for ethical violations related with others-involvement.

Finally, we found a significant correlation between impaired control of mental activities and valence scores for the others-involvement condition. The higher the inhibitory control deficit, the higher perceived pleasure was for ethical violations related with others-involvement. No further relationships were reported (see Table 2 for details).

### 3.4. Relationship between Instrumental Moral Dilemmas and PI Scores

The analysis documents that impaired control of mental activities is positively correlated with acceptability and implicit appraisal, that is, arousal scores associated with dilemmas implying the involvement of others. Therefore, the higher inhibitory control deficit was, the higher the acceptability of ethical violations and respective level of self-reported activation was in response to ethical violations.

We also found a positive relationship between becoming contaminated and the level of arousal associated with dilemmas implying self- and others-involvement. Moreover, arousal was positively correlated with checking behaviors. Therefore, the higher the compulsivity was, the higher the level of self-reported activation was in response to ethical violations.

Finally, a positive correlation was found between urges and worries of losing control over motor behaviors and acceptability. This correlation suggests that acceptability for ethical violations increases with the level of impulsivity. No further relationships were reported (see Table 3 for details).

### 3.5. Relationship between Filler Dilemmas and PI Scores

The analysis documents that arousal positively correlates with checking behavior, impaired control of mental activities, and urges and worries of losing control over motor behaviors. This suggests that self-reported activation (arousal) is predicted by all three dimensions of self-control. Moreover, compulsivity is positively related with acceptability scores. No further relationships were reported (see Table 4 for details).

## 4. Discussion

In the present study we investigated the role played by different dimensions of self-control such as impulsivity, compulsivity, and inhibitory control to predict participants’ moral appraisals for different ethical scenarios [30].

Overall, our results corroborate previous evidence linking self-control with moral judgment [9,36,37]. The involvement of these three dimensions depends on the type of moral scenario and measures (explicit vs. implicit) considered to evaluate the appraisal of participants in response to these scenarios.

A relationship between impulsivity and the explicit measures of moral appraisal was found for incidental and instrumental scenarios. The higher the impulsivity was (measured via the urges and worries of losing control over motor behaviors subscale), the lower the disapproval of ethical violations (i.e., higher acceptability scores). This result was independent of the level of involvement in the ethical dilemmas. Moreover, the greater the impulsivity was, the lower displeasure (i.e., higher valence score) was reported for incidental scenarios related to self-involvement dilemmas. Finally, impulsivity predicted the implicit appraisal (i.e., arousal) for filler scenarios. The greater the impulsivity was, the higher the self-reported arousal rating was.

Compulsivity was found to predict the explicit appraisal of incidental (becoming contaminated) and filler scenarios (checking behaviors). The higher the compulsivity was, the lower the displeasure and acceptability ratings for incidental (self-involvement dilemmas) and filler scenarios were, respectively. Compulsivity was also involved in predicting implicit appraisal for ethical violations related to instrumental and filler scenarios. The greater the compulsive behavior was (measured via the checking-behavior and becoming-contaminated subscales, respectively, for instrumental and filler scenarios) the greater the arousal was. In the case of instrumental scenarios, the correlation with checking behavior was reported only for dilemmas involving the others.

Finally, inhibitory control predicted explicit appraisals for incidental and instrumental scenarios in dilemmas involving others. In particular, the higher the difficulty in the inhibitory control was (measured via impaired control of mental activities subscale), the lower the displeasure and acceptability scores were in response to ethical violations. Moreover, the higher the difficulty in inhibitory control was, the higher the level of arousal was for instrumental (dilemmas involving others) and filler scenarios.

Table 5 provides a summary scheme of the relevance of the three examined self-control dimensions on the examined moral scenarios.

In conclusion, our research provides a novel contribution to research investigating the role of executive functions on ethical reasoning. It suggests that the different dimensions of self-control are involved in moral reasoning, depending on the type of scenario and the considered appraisal (i.e., explicit, or implicit measures). Impulsivity predicts explicit appraisals for incidental and instrumental moral scenarios. Moreover, it predicts the implicit appraisal for filler scenarios; compulsivity predicts explicit appraisal for incidental and filler scenarios, and implicit appraisal for instrumental and filler scenarios. Inhibitory control predicts explicit appraisals for incidental and instrumental scenarios and implicit appraisals for instrumental and filler scenarios.

In conclusion, these data contribute to explain the complex role of self-control on moral decision-making and highlight the importance of taking into consideration the type of moral scenario and respective appraisals, to understand how this executive function may influence ethical behavior.

## 5. Conclusions

This research bears some limitation. First, the numerosity of our sample. A study with a considerably larger sample size is needed to replicate the most pertinent findings of the present research. Second, the absence of autonomic measures that would have provided confirmation of the interpretation of self-reported arousal scores as an implicit measure of moral appraisal. An alternative possibility is that arousal measure considered in our study may be an index of the degree of attention involvement, in line with the evidence of a relationship between arousal and attentional processing in the context of cognitive and behavioral conflicts [38], which probably can also happen while dealing with ethical dilemmas. Third, the absence of a disgust questionnaire. This would have been an interesting variable to include, in line with the evidence of a link between fear of contamination and disgust sensitivity [39,40]. Finally, the disproportion between male and female participants in our sample.

## Figures and Tables

**Table 1 ijerph-18-03840-t001:** Means and standard deviations of Padua Inventory (PI) subscales and of appraisals for the proposed moral dilemmas.

	PI								
	Impaired Control over Mental Activities	Becoming Contaminated	Checking Behaviours	Urges and Worries of Losing Control over Motor Behaviors					
Mean	73,261	24,700	19,317	12,289					
Standard deviation	36,328	15,678	9386	7727					
	**Moral Dilemmas**								
	**Incidental**			**Instrumental**			**Filler**		
	**Acceptability**	**Valence**	**Arousal**	**Acceptability**	**Valence**	**Arousal**	**Acceptability**	**Valence**	**Arousal**
Mean	2018	2373	6275	1876	2512	6266	3140	4616	3595
Standard deviation	1854	2013	2519	2261	2127	2566	2408	2367	2339

PI = The Padua Inventory.

**Table 2 ijerph-18-03840-t002:** Spearman correlation analysis between incidental moral dilemmas ratings and PI scores.

Incidental Dilemmas		Valence	Acceptability	Arousal
Self-involvement	Impaired control of mental activities	Rho = 0.174 *p* = 0.072	Rho = −0.035 *p* = 0.719	Rho = 0.043 *p* = 0.657
	Becoming contaminated	Rho = 0.052 *p* = 0.589	Rho = −0.162 *p* = 0.094	Rho = 0.195 *p* = 0.043 *
	Checking behaviors	Rho = 0.175 *p* = 0.070	Rho = −0.010 *p* = 0.914	Rho = −0.035 *p* = 0.718
	Urges and worries of losing control over motor behaviors	Rho = 0.256 *p* = 0.007 *	Rho = 0.242 *p* = 0.011 *	Rho = −0.101 *p* = 0.297
Others involvement	Impaired control of mental activities	Rho = 0.209 *p* = 0.030 *	Rho = 0.035 *p* = 0.714	Rho = 0.112 *p* = 0.250
	Becoming contaminated	Rho = 0.143 *p* = 0.141	Rho = −0.125 *p* = 0.198	Rho = 0.207 *p* = 0.031 *
	Checking behaviors	Rho = 0.228 *p* = 0.017 *	Rho = −0.078 *p* = 0.422	Rho = 0.049 *p* = 0.615
	Urges and worries of losing control over motor behaviors	Rho = 0.154 *p* = 0.112	Rho = 0.256 *p* = 0.007 *	Rho = −0.009 *p* = 0.924

***** indicates significant results.

**Table 3 ijerph-18-03840-t003:** Spearman correlation analysis between instrumental moral dilemmas ratings and PI scores.

Instrumental Dilemmas		Valence	Acceptability	Arousal
Self-involvement	Impaired control of mental activities	Rho = 0.150 *p* = 0.121	Rho = 0.154 *p* = 0.111	Rho = 0.174 *p* = 0.071
	Becoming contaminate	Rho = −0.026 *p* = 0.785	Rho = 0.082 *p* = 0.396	Rho = 0.191 *p* = 0.047 *
	Checking behaviors	Rho = 0.113 *p* = 0.243	Rho = 0.095 *p* = 0.330	Rho = 0.146 *p* = 0.131
	Urges and worries of losing control over motor behaviors	Rho = 0.169 *p* = 0.081	Rho = 0.148 *p* = 0.128	Rho = 0.007 *p* = 0.938
Others involvement	Impaired control of mental activities	Rho = 0.099 *p* = 0.308	Rho = 0.234 *p* = 0.014 *	Rho = 0.249 *p* = 0.009 *
	Becoming contaminate	Rho = 0.015 *p* = 0.873	Rho = 0.146 *p* = 0.133	Rho = 0.237 *p* = 0.013 *
	Checking behaviors	Rho = 0.026 *p* = 0.785	Rho = 0.149 *p* = 0.125	Rho = 0.230 *p* = 0.016 *
	Urges and worries of losing control over motor behaviors	Rho = 0.154 *p* = 0.111	Rho = 0.226 *p* = 0.019 *	Rho = 0.052 *p* = 0.591

* indicates significant results.

**Table 4 ijerph-18-03840-t004:** Spearman correlation analysis between filler moral dilemmas ratings and PI scores.

Filler Dilemmas	Valence	Acceptability	Arousal
Impaired control of mental activities	Rho = 0.004 *p* = 0.963	Rho = 0.143 *p* = 0.140	Rho = 0.217 *p* = 0.024 *
Becoming contaminated	Rho = 0.168 *p* = 0.082	Rho = 0.214 *p* = 0.026 *	Rho = 0.086 *p* = 0.373
Checking behaviors	Rho = 0.078 *p* = 0.424	Rho = 0.068 *p* = 0.480	Rho = 0.287 *p* = 0.002 *
Urges and worries of losing control over motor behaviors	Rho = −0.002 *p* = 0.976	Rho = 0.145 *p* = 0.134	Rho = 0.225 *p* = 0.019 *

* indicates significant results.

**Table 5 ijerph-18-03840-t005:** Self-control and moral scenarios. The table provides a summary of the predictive role of the examined three dimensions of self-control (impulsivity, compulsivity, inhibitory control), on the different types of moral scenarios (incidental, instrumental, filler) and the respective measures (explicit, implicit).

Dimensions of Self-Control	Incidental		Instrumental		Filler	
	Explicit Measures	Implicit Measures	Explicit Measures	Implicit Measures	Explicit Measures	Implicit Measures
Impulsivity	x		x			x
Compulsivity	x			x	x	x
Inhibitory Control	x		x	x		x

## Data Availability

Data are available by contacting the corresponding author.

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
