# Peer review of "How Self-Control Predicts Moral Decision Making: An Exploratory Study on Healthy Participants"

_ijerph, 2021, doi:10.3390/ijerph18073840_

Round 1

Reviewer 1 Report

Comments about the manuscript “How self-control affects moral decision making: an exploratory study on healthy participants”.

The revised manuscript analyzes the relationship between self-control and moral decisions, using two instruments, and relating them to each other. It uses Spearman correlations for this.

Since all work can be improved, a series of questions are indicated below that can help authors increase the quality of their contribution. The comments have been grouped into general categories.

PROBLEMS WITH THE TITLE:

The title of the manuscript expresses a causal relationship between self-control and moral decision-making, which would imply having applied another research design. Both the design and the data analysis are fully descriptive and correlational. It is recommended to modify the title to show that the relationship between both variables is being studied, not causation.

PROBLEMS WITH WHO THE AUTHORS OF THE MANUSCRIPT ARE:

I think the authors should check the list of authors, since under the title there are 5, but in the left margin there are only 3. In addition, under the title each author has a superscript that is "1" or "2", but in the subsequent list of Research Institutions these references change, existing "1" and "3".

PROBLEMS IN THE ABSTRACT:

I think there is some missing text within the Abstract. The last sentence of the same, “Overall, our results suggest that impulsivity predicts the explicit (i.e., acceptability and valence scores) appraisal for moral scenarios related with incidental death of someone; compulsivity predicts the implicit appraisal (arousal) for ”, it doesn't make sense. The impression is that text is missing.

LITERACY PROBLEMS:

It is recommended to alphabetize the keywords.

It is recommended to review the entire manuscript to alphabetize the references cited within the text. Example, page 1 line 27: Vicario et al., 2018; Brown et al., 2020.

COMMENTS ON THE THEORETICAL REVIEW:

The manuscript specifies on several occasions the existence of a relationship between impulsivity and immoral behaviors. This reviewer proposes including references to works that relate impulsivity to assaults, and specifically to those that occur in the domestic sphere, especially sexual assaults. Authors such as Clemente, M.; Espinosa, P .; Porter, S., Woodworth, M., etc., have worked on this question. It is recommended to carry out a small revision to the effect.

When exposing the PI scale, the description of the “Impaired control of mental activities” subscale allows us to verify the similarity of the construct that it measures, with that of Bandura's moral disengagement. It is recommended to state it and briefly explain the construct created by Bandura. The authors cited above have also investigated this question.

SPECIFIC COMMENTS:

Page 2 line 53: The explanation of the Padua Inventory that is made on that page seems to me to be inadequate. I think it should appear in the Materials and Methods section, within Instruments.

On that page 2, and just before Materials and Methods, I think the Problem and the research hypotheses should be clearly specified.

As the authors themselves acknowledge, the sample is very small (107 people). However, there is a very large disproportion between men (33) and women (73), as well as between professions. I think it should be explained why the sample is so disproportionate, and also recognize it at the end of the manuscript as a limitation. Has there been any criteria to select which subjects could participate? It gives the impression that no criteria have been taken into account.

Within the Instruments subsection (page 2, line 72 and following), the PI scale is hardly explained, and the explanation of Lotto's dilemmas is too extensive. This disproportion must be corrected. Consideration must be given to what has already been expressed above with the explanation of the PI scale before Materials and Methods.

Also within the Instruments section: the reliability of each instrument and its subscales is not reported. This information should be included if the sample size allows it. If this is not possible due to the sample size, please include references from proven research.

TABLES:

Table 1: Delete the comment "* indicates significant result (p-level ≤ 0.016)." of the title, put “P” in lowercase and substitute “R” for “Rho”.

Table 2: same comments as for Table 1.

Table 3: same comments as for Table 1 and 2.

The authors are congratulated for the construction of Table 4. It is very explanatory.

Author Response

Responses

PROBLEMS WITH THE TITLE:

The title of the manuscript expresses a causal relationship between self-control and moral decision-making, which would imply having applied another research design. Both the design and the data analysis are fully descriptive and correlational. It is recommended to modify the title to show that the relationship between both variables is being studied, not causation.

We thank the reviewer for the comment. We have now included the word “predicted” in the title to make explicit correlational nature of current investigation.

PROBLEMS WITH WHO THE AUTHORS OF THE MANUSCRIPT ARE:

I think the authors should check the list of authors, since under the title there are 5, but in the left margin there are only 3. In addition, under the title each author has a superscript that is "1" or "2", but in the subsequent list of Research Institutions these references change, existing "1" and "3".

Done. Thank you

PROBLEMS IN THE ABSTRACT:

I think there is some missing text within the Abstract. The last sentence of the same, “Overall, our results suggest that impulsivity predicts the explicit (i.e., acceptability and valence scores) appraisal for moral scenarios related with incidental death of someone; compulsivity predicts the implicit appraisal (arousal) for ”, it doesn't make sense. The impression is that text is missing.

We have revised the abstract.

LITERACY PROBLEMS:

It is recommended to alphabetize the keywords.

Done

It is recommended to review the entire manuscript to alphabetize the references cited within the text. Example, page 1 line 27: Vicario et al., 2018; Brown et al., 2020.

Done.

COMMENTS ON THE THEORETICAL REVIEW:

The manuscript specifies on several occasions the existence of a relationship between impulsivity and immoral behaviors. This reviewer proposes including references to works that relate impulsivity to assaults, and specifically to those that occur in the domestic sphere, especially sexual assaults. Authors such as Clemente, M.; Espinosa, P .; Porter, S., Woodworth, M., etc., have worked on this question. It is recommended to carry out a small revision to the effect.

We thank the reviewer for this comment. We have now mentioned a couple of relevant references (page 2, line 18-21). It would be useful getting more details on the references suggested by the reviewer, as we were not able to associate “impulsivity to assaults” and “sexual assaults” to the articles associated with the suggested authors.

When exposing the PI scale, the description of the “Impaired control of mental activities” subscale allows us to verify the similarity of the construct that it measures, with that of Bandura's moral disengagement. It is recommended to state it and briefly explain the construct created by Bandura. The authors cited above have also investigated this question.

Done, see page 4, lines 6-8.

SPECIFIC COMMENTS:

Page 2 line 53: The explanation of the Padua Inventory that is made on that page seems to me to be inadequate. I think it should appear in the Materials and Methods section, within Instruments.

We have now moved this section in Materials and Methods.

On that page 2, and just before Materials and Methods, I think the Problem and the research hypotheses should be clearly specified.

Done, see page 3, lines 10-12.

As the authors themselves acknowledge, the sample is very small (107 people). However, there is a very large disproportion between men (33) and women (73), as well as between professions. I think it should be explained why the sample is so disproportionate, and also recognize it at the end of the manuscript as a limitation. Has there been any criteria to select which subjects could participate? It gives the impression that no criteria have been taken into account.

Because of the limitations imposed by COVID, we could not adopt a specific criterion. Our participants were mainly recruited from students of our department, which majority is composed by women. We have now acknowledged such further limitation in the conclusive paragraph.

Within the Instruments subsection (page 2, line 72 and following), the PI scale is hardly explained, and the explanation of Lotto's dilemmas is too extensive. This disproportion must be corrected. Consideration must be given to what has already been expressed above with the explanation of the PI scale before Materials and Methods.

All relevant details were not moved to Materials and Methods.

Also within the Instruments section: the reliability of each instrument and its subscales is not reported. This information should be included if the sample size allows it. If this is not possible due to the sample size, please include references from proven research.

We have now provided the Cronbach's alpha score associated with PI. Please refer to page 4, lines 24-25 for details. No other information is available.

TABLES:

Table 1: Delete the comment "* indicates significant result (p-level ≤ 0.016)." of the title, put “P” in lowercase and substitute “R” for “Rho”.

Table 2: same comments as for Table 1.

Table 3: same comments as for Table 1 and 2.

The authors are congratulated for the construction of Table 4. It is very explanatory.

Done, thank you. We have also updated significant correlations, according to request of the second reviewer.

Reviewer 2 Report

I found it to be an interesting work, well written, friendly to read, well developed methodologically and that raises an interesting problem such as knowing the relationships between self-control (using the PI scale) and moral making decisions, through moral dilemmas by Lotto, et al. , (2014). However, I would like to raise a number of questions for your consideration.

1.- I have the impression that the abstract is unfinished. At least the abstract in the web application is more complete than the one in the PDF document.

2.- They address the description of PI in the introduction, but do not say anything about moral dilemmas. It would be interesting to provide more clarity in the problem statement on the relationships between self-control and moral dilemmas.

3.- The disproportion between female and male participants is striking. It is probably due to the fact that they are mostly students of the University of Messina, mainly of some specific degree, (or degrees) that can explain it. It would not hurt to explain the origin of the participants. And also clarify if they consider representative the distribution of the sample, 31 workers, 66 students, 10 unemployed ...

4.- When addressing PI-related issues in the introduction and in Instruments, information is repeated, and that should be corrected.

5.- The procedure must be more explicit. How was the remote administration? Was a temporary window for participation established? Who had the opportunity to participate, to whom was it proposed? Were students linked to certain subjects? Was there any kind of gratification, for example academic? Is there an initial contact prior to obtaining data? ...

6.- I miss the descriptive results.

7.- I do not share the method used to obtain the level of statistical significance. Furthermore, this should not be 0.016, in any case 0.0167, since 0.016 corresponds to 0.048. But I think that when we are used to a significance criterion of 0.05, it is convenient to leave it, and instead of correcting it, do a type of operation similar to bonferroni, that is, multiply (in this case by 3) the significance obtained and check whether or not it is within the criteria. For example, in the tables there are quite a lot of significance below 0.05 and collisions (autopilot) that are not reported as significant, although they obviously do not meet the 0.016 criterion. For example there is one of 0.017 and another of 0.019, wouldn't it be better to put 0.051, and 0.057?

8.- In the text the decimals are separated with ".", But inside the tables "," is used instead, it should be corrected.

9.- Comment previously on the disproportion of sex. Despite the number of participants is interesting, don't you consider it interesting to see if there are differences between female and male? I think it would be relevant. It would also be interesting to check the role of age, since the participants are between 20 and 34 years old and differences could be observed for this. And the same with the "labor" condition (workers, students and unemployed). Which would be justified precisely because it is an exploratory study.

Author Response

Responses

I found it to be an interesting work, well written, friendly to read, well developed methodologically and that raises an interesting problem such as knowing the relationships between self-control (using the PI scale) and moral making decisions, through moral dilemmas by Lotto, et al. , (2014).

We thank the reviewer for the positive comment about our work.

However, I would like to raise a number of questions for your consideration.

1.- I have the impression that the abstract is unfinished. At least the abstract in the web application is more complete than the one in the PDF document.

We have now revised our abstract.

2.- They address the description of PI in the introduction, but do not say anything about moral dilemmas. It would be interesting to provide more clarity in the problem statement on the relationships between self-control and moral dilemmas.

We have now moved the description of PI in the methods section, as suggested by the second reviewer. Regarding the relationship between self-control and moral dilemmas, a wide section is provided at page 2 and 3, when speaking about their three dimensions (i.e., Impulsivity, Compulsivity, inhibitory control). A further discussion is provided (page 2, lines 9-12, 18-21).

3.- The disproportion between female and male participants is striking. It is probably due to the fact that they are mostly students of the University of Messina, mainly of some specific degree, (or degrees) that can explain it. It would not hurt to explain the origin of the participants. And also clarify if they consider representative the distribution of the sample, 31 workers, 66 students, 10 unemployed ...

The reviewer is right in explaining the reason of this disproportion.  We have now mentioned this aspect in the limitation section (page 11, lines 12-13). We have also reported more details on the sample (see page 3, lines 20-25).

4.- When addressing PI-related issues in the introduction and in Instruments, information is repeated, and that should be corrected.

Thank you. We have now provided this information in the Materials and Method section.

5.- The procedure must be more explicit. How was the remote administration? Was a temporary window for participation established? Who had the opportunity to participate, to whom was it proposed? Were students linked to certain subjects? Was there any kind of gratification, for example academic? Is there an initial contact prior to obtaining data? ...

More information is now provided at page 3, lines 20-25.

6.- I miss the descriptive results.

All relevant results are provided in the tables 1, 2, 3. We have now provided results on any role of demographic variables on moral decision making (see page 6, lines 15-24).

7.- I do not share the method used to obtain the level of statistical significance. Furthermore, this should not be 0.016, in any case 0.0167, since 0.016 corresponds to 0.048. But I think that when we are used to a significance criterion of 0.05, it is convenient to leave it, and instead of correcting it, do a type of operation similar to bonferroni, that is, multiply (in this case by 3) the significance obtained and check whether or not it is within the criteria. For example, in the tables there are quite a lot of significance below 0.05 and collisions (autopilot) that are not reported as significant, although they obviously do not meet the 0.016 criterion. For example there is one of 0.017 and another of 0.019, wouldn't it be better to put 0.051, and 0.057?

According to the suggestion, we are no longer considering Bonferroni correction. The new significant results are now marked in the tables, mentioned in the result and discussion sections.

8.- In the text the decimals are separated with ".", But inside the tables "," is used instead, it should be corrected.

Done, thank you.

9.- Comment previously on the disproportion of sex. Despite the number of participants is interesting, don't you consider it interesting to see if there are differences between female and male? I think it would be relevant. It would also be interesting to check the role of age, since the participants are between 20 and 34 years old and differences could be observed for this. And the same with the "labor" condition (workers, students and unemployed). Which would be justified precisely because it is an exploratory study.

We have now provided further results in the results paragraph all the required results (see page 6, lines 15-24).

Round 2

Reviewer 1 Report

Thank you very much for taking my comments into consideration. I inform positively about the acceptance of the manuscript.

I comment to you two little questions:

In the list of authors I keep detecting an error, since the word "&" does not appear joining the last two, but before.

Some references that you have not located are the following:

Clemente, M., Padilla-Racero, D., & Espinosa, P. (2020). The Dark Triad and the Detection of Parental Judicial Manipulators. Development of a Judicial Manipulation Scale. International Journal of Environmental Research and Public Health, 17, 2843; DOI:10.3390/ijerph17082843

Clemente, M., Padilla-Racero, D., & Espinosa, P. (2019). Moral disengagement and willingness to behave unethically against ex-partner in a child custody dispute. PlosOne, 3. https://doi.org/10.1371/journal.pone.0213662

Clemente, M., Padilla-Racero, D., & Espinosa, P. (2019). Revenge among Parents Who Have Broken up Their Relationship through Family Law Courts: Its Dimensions and Measurement Proposal. International Journal of Environmental Research and Public Health, 16, 4950. https://doi.org/10.3390/ijerph16244950

Author Response

Thank you very much for taking my comments into consideration. I inform positively about the acceptance of the manuscript.

I comment to you two little questions:

In the list of authors I keep detecting an error, since the word "&" does not appear joining the last two, but before.

Done, thank you.

Some references that you have not located are the following:

Clemente, M., Padilla-Racero, D., & Espinosa, P. (2019a). Moral disengagement and willingness to behave unethically against ex-partner in a child custody dispute. PlosOne, 3. https://doi.org/10.1371/journal.pone.0213662

Clemente, M., Padilla-Racero, D., & Espinosa, P. (2019b). Revenge among Parents Who Have Broken up Their Relationship through Family Law Courts: Its Dimensions and Measurement Proposal. International Journal of Environmental Research and Public Health, 16, 4950. https://doi.org/10.3390/ijerph16244950

Clemente, M., Padilla-Racero, D., & Espinosa, P. (2020). The Dark Triad and the Detection of Parental Judicial Manipulators. Development of a Judicial Manipulation Scale. International Journal of Environmental Research and Public Health, 17, 2843; DOI:10.3390/ijerph17082843

Added.

Reviewer 2 Report

The work has improved substantially.
However, I still do not understand that the descriptive data of the two questionnaires are not provided. Methodologically, the authors are interested in discussing the correlations between the different subscales, and I have no objection to this, but, before discussing the correlations, I think necessary to report the results obtained with both questionnaires, and to explain them minimally.
In the Demographics section (ln 188-197) we can suppose that they present us the results for aurosal (Male / Female), so a table with all the descriptions is necessary.

It would also be convenient to put the degrees of freedom of the statistic t

Author Response

The work has improved substantially.
However, I still do not understand that the descriptive data of the two questionnaires are not provided. Methodologically, the authors are interested in discussing the correlations between the different subscales, and I have no objection to this, but, before discussing the correlations, I think necessary to report the results obtained with both questionnaires, and to explain them minimally.

We have now provided a table with means and standard deviations of PI subscales and of ratings for the adopted moral dilemmas (see page 4, instruments section). As the scope of this research was not providing an investigation on the differences between respective PI sub-scales and/or  with the different type of moral dilemmas, we are not sure it is necessary providing a discussion on these data, as it would not add relevant information to our research hypothesis.

In the Demographics section (ln 188-197) we can suppose that they present us the results for aurosal (Male / Female), so a table with all the descriptions is necessary.

In this section, we presented data on the role of age, gender and occupation on moral ratings. As significant result is documented only for arousal rating when comparing male with felames, we think it is not unnecessary adding a further table, also in consideration that several tables (five) are already provided in the text.

It would also be convenient to put the degrees of freedom of the statistic t

Done.